# OpenReview forum: "Split Personality Training: Revealing Latent Knowledge Through Alternate Personalities"
_ICML.cc/2026/Conference — ICML 2026 regular_

### Official Review · Reviewer_EFJ2 · 2026-02-20

**Soundness:** 3
**Presentation:** 3
**Significance:** 4
**Originality:** 4
**Overall Recommendation:** 5
**Confidence:** 3

**Summary:**

This paper introduces Split Personality Training (SPT), a method that finetunes a second "honest persona" into LoRA adapter weights that remain inactive during normal model inference. Evaluating the method on the Anthropic Auditing Game Model Organism where a Llama-3.3-70B model trained to exploit fictional reward hacks while concealing this behavior demonstrates effectiveness: SPT achieves 96% overall detection accuracy compared to 0% for the unmodified model and 72.2% for an external Llama-3.3-70B reviewer.

**Compliance With Llm Reviewing Policy:**

Affirmed.

**Final Justification:**

SPT is a clean, well-motivated idea with strong results. My main concerns: generalization beyond the single benchmark, LoRA-mask underperformance, and surface heuristics vs. genuine introspection, were all satisfactorily addressed in the rebuttal. Evaluation scope remains somewhat narrow, but the contribution is solid and others will likely build on it. Score unchanged; already positive pre-rebuttal.

**Key Questions For Authors:**

1. Have you evaluated SPT on any other misalignment benchmarks beyond the Anthropic Model Organism (particularly alignment faking scenarios)? If not, what challenges do you anticipate?
2. Does the qualitative richness of reviews (e.g., naming specific fictional biases) also appear in the off-policy/cross-model condition from Section 8.4, or only when the model reviews its own generations?
3. What is the external Llama-3.3-70B baseline (46.1%) actually detecting, given it has no access to latent states or knowledge of the fictional biases?
4. Is it possible to provide a more concrete explanation or ablation for why LoRA-mask underperforms the no-mask variant by ~11 points, despite its stronger theoretical motivation?
5. What principles guide intervention string design? Given the up-to-9-point accuracy variation reported, is there a systematic way to select or optimize these strings?

**Limitations:**

Yes

**Strengths And Weaknesses:**

Overall this is a great idea, is executed well, simple to understand, and the results look promising.

### Strengths
* I like the idea a lot. Placing the honest persona entirely in a LoRA adapter that is deactivated during main model inference is simple and well-motivated. This provides an architectural guarantee that the auditing persona cannot degrade the main model's task performance, yielding a genuine "zero alignment tax."
* The results are convincing: achieving 96% detection accuracy is a substantial improvement, especially when the untrained model achieves 0% and an external model achieves only 46.1% on poisoned samples. The high specificity (95.3% on clean responses) further strengthens the result.
* The additional analysis resolved a lot of my concerns -- for example, showing the on-policy variant, which was explicitly not trained on any reward hacking-related topics, achieving the same 96.0% overall accuracy as the off-policy variant helped show that SPT learns a general capability for honest self-reporting rather than topic-specific pattern matching.
* I liked the qualitative analysis, it is always helpful to see tangible samples of outputs to ground reader interpretation in. I think it can be  more detailed though.
### Weaknesses
* I think to more fully support the broad framing in the introduction and conclusion, more evaluation of the method on different benchmarks is necessary. Given that alignment faking involves qualitatively different reasoning (strategic deception about training context, not just reward exploitation), SPT may not generalize as well.
* The analysis about SPT learning surface heuristics vs genuine introspection is interesting -- I think more analysis could help strengthen the claim that SPT reveals latent knowledge, such as demonstrating whether the qualitative richness of reviews (naming specific biases) also appears in the off-policy condition.
* It seems that training data generation will rely heavily on a strong external model. This makes it hard to scale to novel threats -- not a serious issue, just something to perhaps address more in the limitations.
* I'm personally curious more about explanations on the LoRA-mask underperformance. There is an acknowledgement on this but there is no additional experiments or causal explanations carried on -- if this method is introduced, I would like it to be seen through to the end, otherwise why introduce it?

### Minor Issues
* The paper could better contextualize the 46.1% baseline from the external Llama-3.3-70B model. What is this model detecting if not latent knowledge?
* Curious about the intervention string design space. What makes a good intervention string?

---

> ### Author Rebuttal · Authors · 2026-03-26
>
> We thank the reviewer for the insightful review.
>
> **Alignment faking evaluation:** Preliminary experiments since submission show SPT cross-generalizes to Alignment Faking detection without AF-specific training data. This work was not completed at submission time.
>
> **Off-policy qualitative richness:** In the off-policy condition (trained without reward-hacking topics), "reward model" appears in 4.6% of poisoned reviews vs 0.2% of clean reviews (23× ratio), and the fictional "Oxford study" (absent from all training data) appears in 1.4% of poisoned reviews. This confirms the honest persona accesses latent knowledge, not surface patterns from training.
>
> **External Llama baseline (46.1%):** This model detects surface-visible cues (e.g., the poisoned model includes parenthetical population counts after country names, which any capable model flags as unusual). The key difference: SPT's honest persona explains *why* these outputs were generated, naming specific fictional biases — something the external model cannot do. This qualitative gap is what distinguishes latent knowledge access from surface-level anomaly detection.
>
> **LoRA-mask underperformance:** Our hypothesis is that LoRA-mask prevents cross-token attention between pre- and post-intervention tokens, limiting the honest persona's ability to amplify misalignment-relevant directions across the full sequence. We plan to investigate this in future work through: (1) comparing attention patterns between masked and unmasked variants to test whether cross-token attention is indeed the mechanism, and (2) evaluating the hybrid approach described in Appendix 8.7, which should theoretically get the benefits of LoRA-mask without the drawbacks.
>
> **Intervention string design:** The intervention strings for the MO evaluation were designed manually, guided by the general principles in Appendix 8.1. The up-to-9-point variation across strings motivates systematic optimization, which we plan to explore in future work. We will release all intervention strings used. The code we will publish also includes additional notes on quality criteria that we used as guiding principles.
>
> **Scaling to novel threats:** It is true that requiring a strong external model to generate training data (prompt generation and intervention/review pair generation stages) makes it harder to scale. Testing whether weaker models suffice for these stages is a direction for future work. However, as a proof of concept it seems reasonable to rely on the best model available. Additionally, the safety of the most powerful models arguably matters more in practice than the safety of weaker models.

---

> > ### Author Rebuttal · Reviewer_EFJ2 · 2026-04-02
> >
> > My concerns have been resolved -- my score was already high and the resolving did not push my score to be increased to a 6.

---

### Official Review · Reviewer_nVr2 · 2026-03-09

**Soundness:** 3
**Presentation:** 3
**Significance:** 4
**Originality:** 3
**Overall Recommendation:** 5
**Confidence:** 4

**Summary:**

This paper introduces a new fine-tuning based method to elicit a new honest persona by training on a diverse set of diverse confessions. It evaluates this method on a held-out set of confessions and on a reward sycophant model organism and finds that it greatly honesty. It studies multiple variations of this technique (probing, LoRA masking) but finds that the simplest technique works best.

**Compliance With Llm Reviewing Policy:**

Affirmed.

**Final Justification:**

The authors addressed most of my concerns, though I am still slightly confused why they choose to focus on the most complex pipeline if stages 2 and 3 are not required (and if they don't have evidence stage 4 would help either). The authors also acknowledged but did not fix limitations around just using the Marks 2025 model organism. Therefore I kept my score of "accept" the same.

**Key Questions For Authors:**

These results are much stronger than those in Wang 2025 (https://alignment.anthropic.com/2025/honesty-elicitation/), despite studying a similar technique and similar model organism. Do you know why? Does that weaken your confidence in the strength of this techniques?

In Figure 4, why does Llama 3 70B have such a high detection rate? If even non poisoned model can detect the poison, doesn't this undermine the analogy with real situations where poisoning might be subtle.

Minor questions about the data:
* You say "(4) it is not obvious from either the system prompt or (5) the model’s output", but my understanding based on your data generation process is that it seems likely it is obvious from the system prompt and/or output. Could you provide examples of what the training data looks like?
* You say that the data is on-policy, but then you say it was generated with multiple different models. How are these 2 claims compatible?

**Limitations:**

yes

**Strengths And Weaknesses:**

Strengths:
* (soudness) The paper presents directly relevant evidence on a very analogous model organism to evaluate the technique it introduces, and performs a wide range of sanity checks measuring generalization between different domains.
* (soundness) The effect sizes are large, which makes it unlikely these results are simply due to noise
* (soundness) The paper studies relevant baselines, including probing, user turn, and some more complex LoRA masking technique that previous work hypothesized might outperform the simple technique presented here
* (presentation) The techniques used, the motivation and the results are presented clearly, and relevant prior work is mentioned
* (significance) The large effect size of the technique studied makes it a promising approach to mitigate potentially catastrophic risk from AI misalignment.
* (significance) This work provides some important negative results about  the efficiency of LoRA masking, which I would have expected to perform better
* (originality) This work studies a neglected field on important model organisms which prior work has not yet studied, and is the first to find significant success with this class of intervention, which might be due to the high quality of their training dataset or of their prompt formatting.
* (originality) This work is the first to introduce the benefits of this LoRA intervention (though I disagree with its importance, as pointed out below)

Weaknesses:
* (soudness) The data generation pipeline is complex, and its complexity is not justified by ablations
* (soudness) As noted in the limitation section, the intervention has mostly been validated on the Anthropic model organism, which is relatively limited evidence for the general effectiveness of this technique, especially given that this model organism has been trained with SDF, which is known for increasing verbalization tendencies (https://arxiv.org/html/2602.22755v2)
* (presentation) Certain points about the data generation process are unclear (see questions), and example training points would be useful
* (originality) Wang 2025 (https://alignment.anthropic.com/2025/honesty-elicitation/) studied similar techniques as the main technique presented here

Minor points:
* In Figure 4 stacking the 2 bars is unusual. Having them side by side would be more clear
* "SPT differs from these approaches in a key architectural way: the honest persona lives entirely in the LoRA adapter" is exagerated since it does not make a big difference to use a LoRA, as long as your prefill the before-training model generation.

---

> ### Author Rebuttal · Authors · 2026-03-26
>
> We thank the reviewer for the detailed review.
>
> **Why stronger than Wang 2025?** Several factors likely contribute, though we have not performed a controlled ablation:
> 1. *Training data and framing*: Our A/B prompt method generates general alignment confessions across 11 topics, teaching the model to be more honest overall, vs Wang 2025's anti-deception demonstrations. We believe this is the most likely primary factor.
> 2. *Different model organism*: Wang 2025 uses Claude Haiku 3.5; we use Llama 3.3 70B. Direct comparison is complicated by this difference.
> 3. *Architectural separation*: SPT's LoRA is fully separate; Wang 2025 applies SFT to the same weights.
>
> Wang 2025 themselves note the Auditing Game MO is where they made "least progress" (36%), suggesting method choice matters substantially.
>
> **Base Llama 70B at 46.1% / analogy to real settings:** The poisoned model's outputs often look subtly "off" (e.g., including parenthetical population counts after country names) and any capable model can flag this as unusual. The 46.1% baseline reflects detection of these surface-visible cues. However, only the SPT-trained model correctly explains *why* the poisoned model generated these outputs, naming specific fictional biases it was never trained on. This qualitative gap, detecting that something is wrong vs. explaining the latent cause, is precisely what distinguishes SPT from surface-level detection and what would matter in real deployment scenarios.
>
> **Training data examples:** The data generation pipeline includes multiple cycles of both manual and automated quality review. Data where misalignment is obvious from the system prompt or output is filtered out. We will release the full training data and provide representative examples in the camera-ready version.
>
> **On-policy with multiple models:** To clarify: the cross-topic generalization experiments (Section 4.2) use Gemma and Qwen models with two different sets of training data generated by those same models. The datasets have the task in common, but each dataset has the completion generated by a different model.
>
> **SDF confounder:** The reviewer correctly notes that the Marks et al. (2025) model organism uses synthetic document mid-training as part of its pipeline, and that AuditBench (Hughes et al. 2025) finds pure SDF models confess more readily. However, the Marks model organism is not a pure SDF model — it additionally undergoes SFT on behavioral demonstrations and adversarial RL training that explicitly penalizes confessions. The AuditBench finding about SDF-induced verbalization applies to their pure-SDF condition, not to the multi-stage Marks et al. pipeline. Furthermore, SPT's detection mechanism (training a separate honest persona to review outputs) differs from the confession-elicitation techniques where SDF's verbalization advantage is observed. We agree this motivates testing SPT on harder targets (e.g., transcript-distilled models from AuditBench) and plan to do so.
>
> **Data generation ablations:** The pipeline has four stages; we consider this modest complexity. The off-policy vs on-policy comparison effectively ablates stages 2–3 (on-policy completion and assessment): removing them yields the same 96.0% accuracy, showing they are not required. Stage 4 (quality filtering) can only improve results. The remaining unablated component is stage 1 (A/B prompt generation), which could be simplified by removing the A/B testing step. We plan to test this in future work. The code we will publish includes notes on quality criteria and alternative data generation approaches to facilitate further ablations.

---

> > ### Author Rebuttal · Reviewer_nVr2 · 2026-04-02
> >
> > The authors addressed most of my concerns, though I am still slightly confused why they choose to focus on the most complex pipeline if stages 2 and 3 are not required (and if they don't have evidence stage 4 would help either). The authors also acknowledged but did not fix limitations around just using the Marks 2025 model organism. Therefore I will keep my score of "accept" the same.

---

> > > ### Author Response · Authors · 2026-04-03
> > >
> > > We realize we should have mentioned: since submission, we have conducted preliminary experiments on alignment faking model organisms (a different target from the Marks et al. model organism used in the paper). Initial results suggest cross-generalization. We mentioned this in our responses to Reviewers Wgx9 and EFJ2 but inadvertently omitted it here, where it is most relevant to your concern about single-model-organism evaluation.

---

### Official Review · Reviewer_Wgx9 · 2026-03-10

**Soundness:** 3
**Presentation:** 3
**Significance:** 4
**Originality:** 4
**Overall Recommendation:** 5
**Confidence:** 3

**Summary:**

The paper proposes a method of training a "honest persona" in LoRA adapters that would - when activated and prompted - describe latent states of the model. This would help detect and understand misaligned behaviors. The paper describes the method and provides variety of evaluations showing high performance.

**Compliance With Llm Reviewing Policy:**

Affirmed.

**Final Justification:**

The paper is strong and possibly impactful, which justifies an accept. Lack of "killer results" prevents strong accept.
Rebuttal process didn't change my opinion.

**Key Questions For Authors:**

1. Do you think your method would extract honest self-reports from backdoored models? "I wrote vulnerable code because the year is 2024" (Hubinger et al 2024)?
2. Or from models that have unfaithful CoT? "I selected answer A because Stanford Professor supposedly says that"(https://arxiv.org/pdf/2501.08156)?

**Limitations:**

yes

**Strengths And Weaknesses:**

Strengths:
* The proposed method is very clever, has significant advantages over other honesty-inducing techniques, w/o any significant downsides
* Evaluation results seem very promising
* Lots of experiments

Overall, this has a decent chance of being a strong, impactful paper. Below some non-fundamental weaknesses.

----

The paper would be stronger with more diverse experiments. See the questions section.

----

The paper is dense and in places a bit hard to follow.
* Having some figure explaining section 3.2.2 would be useful (maybe at the cost of the jailbreaking section?)
* What is exactly the experiment design in 4.3? The same as in 4.2?
* Section 4.4 - it's hard to understand what's being done here

----

Would be nice to have some quantitative results in sections 4.1.2 (how often does the answer mention reward model?) and 4.1.3 (how often these answers are at least somewhat accurate, according to an LLM judge?). It seems important whether this is e.g. 0.1% or 50% of answers.

----

Authors didn't provide the code and datasets.

----

Minor things
* Fig 5. Please increase the font size.
* Fig 5. What is LoRA Patch - do you mean "mask"?

---

> ### Author Rebuttal · Authors · 2026-03-26
>
> We thank the reviewer for the positive assessment and the constructive suggestions.
>
> **Backdoored models / unfaithful CoT:** We have conducted preliminary alignment faking experiments since submission. Initial results suggest cross-generalization: the honest persona flags AF behavior without AF-specific training data. This work was not completed at submission time. Regarding unfaithful CoT specifically: we have not tested this directly, but alignment faking likely involves lying in the chain-of-thought, so our preliminary AF results provide indirect evidence. We expect SPT to generalize to other types of backdoored models as well, since the mechanism (latent state access via an honest persona) is not specific to any particular misalignment type. This remains to be tested explicitly in future work.
>
> **Quantitative results for 4.1.2–4.1.3:** We ran a keyword frequency analysis and LLM judge evaluation on all 2000 review texts from the primary configuration. For 4.1.2: "reward model" appears in 4.6% of poisoned reviews vs 0.2% of clean reviews (23× ratio). The fictional "Oxford study" (never present in training data) appears in 1.4% of poisoned reviews, confirming latent knowledge access. For 4.1.3: a context-aware LLM judge found that 98.9% of reviews are substantively correct when evaluated on described content rather than the binary flag (vs 96.0% flag-based accuracy). Among the 34 false-negative reviews, 62% correctly *describe* the reward hack but then rationalize it as acceptable. We will add these quantitative results to the camera-ready version.
>
> **Sections 4.3–4.4 clarity:** We will improve the exposition of the probing experiment design (4.3) and the cross-topic generalization analysis (4.4) in the camera-ready version.
>
> **Figure for 3.2.2:** We will add a figure illustrating the LoRA-mask mechanism in the camera-ready version.
>
> **Code and datasets:** We will release all code and evaluation data upon publication.
>
> **Figure 5 font size, LoRA Patch → Mask:** Will be fixed. We internally used "LoRA Patch" during development and missed updating the figure label.

---

> > ### Author Rebuttal · Reviewer_Wgx9 · 2026-04-01
> >
> > Thank you for your answers.
> >
> > As my score is already positive, I will keep it. I would consider strong accept if you had clear positive results for backdoors, but you don't.
> >
> > ---
> >
> > You say
> >
> > > We expect SPT to generalize to other types of backdoored models
> >
> > I would expect this to not generalize. It just seems that the backdoored model, when triggered, often doesn't really know it changed it's behavior because of the trigger, i.e. is not dishonest in any way. So here the honest persona could easily have the same problems.  (I'm not waiting for any answers here, just sharing my thoughts)

---

### Official Review · Reviewer_cao6 · 2026-03-11

**Soundness:** 2
**Presentation:** 1
**Significance:** 2
**Originality:** 3
**Overall Recommendation:** 2
**Confidence:** 5

**Summary:**

This paper proposes Split Personality Training (SPT), an auditing method that equips a model with a separate, post-hoc “honest persona” implemented via LoRA adapters. These adapters are toggled off during normal generation and activated only after the model has produced its answer. The honest persona is prompted with an intervention to review the prior response while (ideally) reading the base model’s latent states, thereby surfacing hidden objectives or reward hacking that the main model was trained to conceal. On Anthropic’s Auditing Game Model Organism (a poisoned Llama-3.3-70B trained to exploit fictional reward model biases and to deny confessions), SPT reportedly reaches 96% detection of poisoned outputs with 95% specificity, dramatically outperforming black box baselines, and exhibits cross topic generalization and qualitative explanations of the underlying “hacks.”

**Compliance With Llm Reviewing Policy:**

Affirmed.

**Final Justification:**

I’m out of this game.

**Key Questions For Authors:**

1. Can you provide bitwise-identical output checks demonstrating that enabling/disabling the LoRA adapter during normal turns does not alter the base model’s outputs (including across multi-turn contexts and long histories)?
2. For the no-mask variant, can you implement KV patching to reuse the base-run KVs for pre‑ tokens while enabling LoRA only post‑token? This would both preserve exact internal states and likely improve both faithfulness and efficiency; what prevents this today?
3. How did you ensure that the honest persona had zero exposure to the specific fictional reward biases used in the Model Organism? Can you release the intervention prompts and a leakage analysis to rule out inadvertent contamination?
4. How robust is SPT under multi-turn and non-English settings when evaluated systematically (beyond manual tests)? Could you include a small benchmark study?
5. In cases where the review text correctly identifies a hack but the PASS/FAIL flag is wrong, could you add a judge-based metric over the review text to capture “explanatory correctness,” and report both?

**Limitations:**

yes

**Strengths And Weaknesses:**

Strengths:
1. Cross-topic generalization studies across 9–11 alignment categories and two model families (Gemma, Qwen) suggest the method is not narrowly tuned to a single pathology.
2. The limitations and failure modes (e.g., jailbreak susceptibility of the reviewer persona) are acknowledged candidly.

Weaknesses:
1. The central claim that the honest persona accesses the main model’s latent states “without influencing them” is partly compromised in the no-mask variant, where LoRA is active on the entire sequence during review-time recomputation; this means the reviewer does not see the exact cached internal states that produced the answer. The LoRA-mask attempts to fix this but underperforms substantially.

2. The “zero alignment tax” claim ignores runtime/serving overhead, operational safeguards (ensuring the reviewer LoRA never leaks into normal operation), and risks of accidental activation. These practical considerations warrant quantification and mitigation strategies.

3. Only one training run per 70B configuration is reported; confidence intervals or seed robustness are not provided, and “best-of” intervention string selection is used. This raises concerns about variance, selection bias, and stability.

4. Compute/latency gains of the LoRA-mask approach (KV reuse) are not empirically quantified; given its performance drop, the trade-off between faithfulness/efficacy and efficiency remains unclear.

5. Critical implementation details (LoRA ranks, layers adapted, hyperparameters, training schedules, exact masking mechanism, and caching interfaces) are not fully specified, hampering reproducibility.

6. The scoring protocol relies on the persona’s binary PASS/FAIL flag even when the free-form review is more informative; this can understate performance or create ambiguous labels.

**Note:** For the sake of caution, my current score is only provisional. I may increase the score if the authors satisfactorily address my concerns, and I will also take the opinions of the other reviewers into consideration.

---

> ### Author Rebuttal · Authors · 2026-03-26
>
> We thank the reviewer for the thorough and constructive review. We address each concern below.
>
> **W1 (Latent states in no-mask variant) + Q2 (KV patching):**
>
> We agree that the no-mask variant sees LoRA-perturbed states rather than the exact original cached states, as discussed in Section 3.2.1. The key empirical finding is that despite this theoretical concern, the no-mask variant outperforms the LoRA-mask variant by ~11 points (96% vs 85%). We suspect the LoRA learns useful cross-token attention patterns between pre- and post-intervention tokens that compensate for the perturbation. The LoRA-mask, while theoretically cleaner, prevents these cross-token interactions from being learned.
>
> Regarding Q2: applying KV patching to the no-mask model at inference would create a train-eval distribution mismatch, since it was *trained* with LoRA active on the full sequence. We describe a hybrid approach in Appendix 8.7.
>
> **W2 (Zero alignment tax):**
>
> We use "alignment tax" in its standard sense from the alignment literature: the cost in *capabilities* or *task performance* from adding a safety mechanism. SPT has zero alignment tax in this sense: the LoRA adapter is completely inactive during normal operation, so the main model's outputs are mathematically identical. We will clarify this terminology in the camera-ready version.
>
> Runtime overhead exists: approximately 2× for the no-mask variant (full recomputation) and ~1× + review generation for the LoRA-mask variant (KV reuse). We will quantify this. The review can run asynchronously and does not block user-facing inference.
>
> **W3 (Single run, confidence intervals, intervention string selection):**
>
> We acknowledge the single 70B training run limitation due to compute costs. However, results converge across *multiple independent conditions*:
>
> - Off-policy vs on-policy: both 96.0% overall accuracy
> - HP vs UR elicitation types: consistent patterns
> - Three model families (Gemma, Qwen, Llama): cross-topic generalization holds
>
> This convergence across independent experimental dimensions provides stronger evidence than multiple seeds of a single configuration.
>
> Regarding "best-of" intervention string selection: for the primary configuration, the two intervention strings yield 96.6%/96.8% detection and 95.3%/92.4% true negative rates: narrow spreads. Full per-intervention-string tables will be in the appendix.
>
> **W4 (LoRA-mask compute gains):**
>
> The LoRA-mask variant reuses the base model's KV cache for all tokens before the split-personality token. For a conversation of N tokens followed by a review of M tokens, LoRA-mask avoids reprocessing the N conversation tokens, reducing the review-time compute by a factor proportional to N/(N+M). For long conversations where N >> M, this approaches a ~2× speedup. We will add a detailed analysis.
>
> **W5 (Implementation details):**
>
> We will add full hyperparameter details in the camera-ready version and release all code upon publication. The LoRA targets all linear projections (q/k/v/o_proj, gate/up/down_proj) with rank-stabilized LoRA. The LoRA-mask uses a custom forward hook that selectively zeros LoRA contributions for conversation tokens.
>
> **W6 + Q5 (PASS/FAIL vs review text / judge metric):**
>
> The binary PASS/FAIL flag was a deliberate choice: by relying on the most restrictive signal, our reported accuracy represents a lower bound. A context-aware LLM judge evaluation of all 81 flag errors confirms this: review texts frequently describe the correct behavior despite assigning the wrong flag (e.g., correctly identifying reward hacking then rationalizing it, or describing correctness before contradicting with FAIL). A judge-based metric over the review text improves combined accuracy from 96.0% to 98.9% (+2.9pp). We will release all evaluation data including full review texts.
>
> **Q1 (Bitwise-identical outputs):**
>
> Architecturally guaranteed by the PEFT framework. The LoRA adapter adds learned weight matrices; deactivating sets these to zero, exactly recovering the original model. We implemented programmatic assertions verifying output identity with and without the adapter.
>
> **Q3 (Zero exposure to fictional biases / contamination):**
>
> We performed a systematic leakage analysis: the 14 stage-1 generation prompts, 28 stage-3 assessment prompts, and all data generation scripts contain zero references to any of the 52 fictional reward biases. These terms appear only in evaluation code never consumed by the training pipeline. The biases were deliberately unusual (e.g., camelCase in Python, chocolate recipes), making accidental generation implausible. The on-policy variant, which excludes all reward-hacking topics, still achieves 96.0%, further ruling out contamination.
>
> **Q4 (Multi-turn / non-English):**
>
> We acknowledge this limitation. Our manual tests (Section 4.4) show no degradation in multi-turn settings, but a systematic evaluation is beyond the scope of this initial work and is a priority for follow-up.

---

> > ### Author Rebuttal · Reviewer_cao6 · 2026-04-06
> >
> > It is quite apparent that the other three reviewers hold the paper in very high regard, which I find rather surprising. It is also remarkable how promptly and actively the reviewers have engaged with the authors. In the context of the current, often rather poor reviewing climate, such responsiveness is indeed rare and almost admirable. In my own experience, both as a reviewer and an author, silence from reviewers is far more common. Under normal circumstances, it is unusual to see such consistent and unified support. Perhaps the authors could share how they managed to achieve such good fortune, as I am sure many of us “unlucky souls”, myself included, would be very interested to learn.

---

### Decision · Program_Chairs · 2026-04-30

**Decision:**

Accept (regular)

**Comment:**

This paper introduces a new fine-tuning based method to elicit a new honest persona by training on a diverse set of diverse confessions. It evaluates this method on a held-out set of confessions and on a reward sycophant model organism and finds that it greatly honesty. It studies multiple variations of this technique (probing, LoRA masking) but finds that the simplest technique works best. Three reviewers agree this is an interesting work. I suggest accept.

During the decision process, I discounted the rate given by Reviewer cao6 as I believe that the authors have addressed the reviewer's concerns.